# The Usefulness of Questionnaires in Assessing the Risk of Obstructive Sleep Apnea in Patients in the Managed Care after Acute Myocardial Infarction Program—The Results of a Cross-Sectional Study

**DOI:** 10.3390/jpm13040642

**Published:** 2023-04-07

**Authors:** Danuta Łoboda, Michalina Stepanik, Anna Szajerska-Kurasiewicz, Maciej Turski, Dagmara Urbanczyk-Świć, Jacek Durmała, Krzysztof S. Gołba

**Affiliations:** 1Department of Electrocardiology and Heart Failure, Medical University of Silesia in Katowice, 40-635 Katowice, Poland; 2Department of Electrocardiology, Upper-Silesian Medical Centre, 40-635 Katowice, Poland; 3Department of Rehabilitation, Medical University of Silesia in Katowice, 40-635 Katowice, Poland; 4Day Treatment Cardiac Rehabilitation Ward, Upper-Silesian Medical Centre, 40-635 Katowice, Poland

**Keywords:** cardiac rehabilitation, myocardial infarction, obstructive sleep apnea, risk prediction

## Abstract

Obstructive sleep apnea (OSA) worsens prognosis after myocardial infarction (MI) but often remains undiagnosed. The study aimed to evaluate the usefulness of questionnaires in assessing the risk of OSA in patients participating in managed care after an acute myocardial infarction program. Study group: 438 patients (349 (79.7%) men) aged 59.92 ± 10.92, hospitalized in the day treatment cardiac rehabilitation department 7–28 days after MI. OSA risk assessment: A 4-variable screening tool (4-V), STOP-BANG questionnaire, Epworth sleepiness scale (ESS), and adjusted neck circumference (ANC). The home sleep apnea testing (HSAT) was performed on 275 participants. Based on four scales, a high risk of OSA was found in 283 (64.6%) responders, including 248 (56.6%) based on STOP-BANG, 163 (37.5%) based on ANC, 115 (26.3%) based on 4-V, and 45 (10.3%) based on ESS. OSA was confirmed in 186 (68.0%) participants: mild in 85 (30.9%), moderate in 53 (19.3%), and severe in 48 (17.5%). The questionnaires’ sensitivity and specificity in predicting moderate-to-severe OSA were: for STOP-BANG—79.21% (95% confidence interval; CI 70.0–86.6) and 35.67% (95% CI 28.2–43.7); ANC—61.39% (95% CI 51.2–70.9) and 61.15% (95% CI 53.1–68.8); 4-V—45.54% (95% CI 35.6–55.8) and 68.79% (95% CI 60.9–75.9); ESS—16.83% (95% CI 10.1–25.6) and 87.90% (95% CI 81.7–92.6). OSA is common in post-MI patients. The ANC most accurately estimates the risk of OSA eligible for positive airway pressure therapy. The sensitivity of the ESS in the post-MI population is insufficient and limits this scale’s usefulness in risk assessment and qualification for treatment.

## 1. Introduction

Untreated obstructive sleep apnea (OSA) worsens the prognosis of patients with cardiovascular (CV) diseases by multiplying the risk of developing coronary artery disease (CAD), heart failure, ventricular and supraventricular arrhythmias, resistant hypertension, stroke, and metabolic dysfunction [1,2,3,4,5,6,7,8,9]. OSA significantly increases the risk of first and subsequent coronary events and CV death after myocardial infarction (MI), and the risk of major adverse cardiovascular events after percutaneous coronary interventions [9,10,11,12]. OSA increases pulmonary morbidity and prolongs hospitalization after coronary artery bypass grafting [13]. The presence of OSA in post-MI patients is associated with adverse left ventricular (LV) remodeling, systolic LV dysfunction, and an increased risk of ventricular arrhythmias and sudden cardiac death [5,14]. Resistant hypertension in OSA leads to LV hypertrophy, LV diastolic dysfunction, and episodes of atrial fibrillation [15], which impairs patients’ exercise capacity and increases the risk of complications associated with combined antiplatelet and antithrombotic therapy [16]. OSA, arterial hypertension, chronic kidney disease, chronic obstructive pulmonary disease, and primarily, obesity with concomitant metabolic disorders or diabetes contribute to the development of heart failure with preserved ejection fraction with coronary microvascular dysfunction, myopathy, and atrial and ventricular fibrosis [17,18]. It is known that OSA reduces exercise tolerance [19,20], which may affect the effectiveness of cardiac rehabilitation (CR) after MI. Moreover, sleep fragmentation in OSA patients is associated with many troublesome symptoms, such as daytime sleepiness, irritability, memory deterioration, and depression, thus worsening the quality of life [21] and compliance in post-MI patients.

OSA-related CV diseases’ basic mechanisms are: (1) intermittent hypoxia/reoxygenation, oxidative stress, and inflammation affecting endothelial function, and participate in plaque instability/plaque susceptibility and arterial stiffness [8,22,23]; (2) metabolic dysfunction with insulin resistance accelerating coronary microvascular dysfunction and coronary and peripheral atherosclerosis [6,18,23]; (3) negative intrathoracic pressure during respiration with closed airways with a repetitive increase in LV wall tension and subsequent septal hypertrophy/LV concentric hypertrophy, or LV remodeling with chamber dilatation [14,15,17,24,25]; (4) impaired coronary perfusion during increased LV transmural pressure, increased afterload, and OSA-related hypoxemia, resulting in nocturnal myocardial ischemia [25,26]; and (5) activation of the sympathetic nervous system and the renin-angiotensin-aldosterone system with peripheral vasoconstriction, sodium and water retention, and resistant hypertension [3,25,26].

Overweight/obesity, high blood pressure, diabetes, hyperlipidemia, and smoking—the main risk factors for OSA and CAD, including the male sex, age over 35—are similar [27]. Despite this, many patients with MI remain undiagnosed with OSA [7,28], whose gold standard for diagnosis is polysomnography [29]. However, implementing simple OSA risk questionnaires into routine cardiac diagnostics may improve cardiologists’ cooperation with sleep or respiratory physicians in tackling a common threat [30].

The study aimed to evaluate questionnaires’ usefulness in estimating the risk of moderate-to-severe OSA in patients participating in the Polish Cardiac Society, the National Health Fund, and the Ministry of Health comprehensive managed care after acute myocardial infarction program (MC-AMI, KOS-ZAWAL).

## 2. Materials and Methods

### 2.1. Study Design and Patients

The study was cross-sectional and included consecutive patients participating in the MC-AMI program who were hospitalized in the Day Treatment Cardiac Rehabilitation Department of the Upper-Silesian Medical Centre (Katowice, Poland). The inclusion criteria for the study was a history of MI in the 7 to 28 days before admission and were consistent with the inclusion criteria for the MC-AMI program, described in detail elsewhere [31,32]. In brief, the MC-AMI program covers all patients hospitalized for acute MI (both ST-elevation MI and non-ST-elevation MI) diagnosed in line with the fourth universal definition of myocardial infarction [33]. The MC-AMI program provides treatment in four treatment modules: Module I—the treatment of the acute phase of MI (i.e., angioplasty, arterial bypass grafting, follow-up visit within 14 days after discharge); Module II—cardiac rehabilitation; Module III—electrotherapy (i.e., implantation of an implantable cardiac defibrillator in a primary prevention of sudden cardiac death, if necessary); and Module IV—specialized cardiac care during the 12 months following MI. The study exclusion criterion was the current treatment of OSA with positive airway pressure or intraoral devices.

We estimated the prevalence of OSA risk factors based on anthropometric measurements, medical history and scores on OSA risk scales in the entire study group and subgroups according to sex and the severity of OSA as determined by the home sleep apnea test (HSAT, portable level 3 sleep test). Then, we assessed the sensitivity, specificity, and positive and negative predictive values of the questionnaires in estimating the risk of moderate-to-severe OSA. Patients with HSAT-proven central or mixed sleep apnea were excluded from this analysis.

### 2.2. OSA Risk Assessment

The risk of OSA was assessed as a part of the routine medical history with regard to hospital admissions by standardized risk scales or questionnaires [29,34,35,36,37,38]: the STOP-BANG, the 4-variable screening tool (4-V), the Epworth sleepiness scale (ESS), and the adjusted neck circumference (ANC), as shown in Table 1. The following results indicated a high risk of OSA: (1) STOP-BANG ≥ 5 points or ≥2 points (snoring, tiredness, observed apnea, or high blood pressure) plus male sex, or body mass index > 35 kg/m^2^, or neck circumference > 40 cm; (2) 4-V > 13 points; (3) ESS > 10 points; and (4) ANC > 48 cm.

### 2.3. Polysomnography Evaluation

All participants who provided informed consent for further evaluation—irrespective of the estimated OSA risk—underwent the HSAT, using an Alice NightOne (Philips Respironics) device [29]. The HSAT recordings were analyzed following the recommendations of the American Academy of Sleep Medicine [29,39]. Sleep apnea was defined as the airflow’s complete cessation or reduction by ≥90% for ≥10 s through the respiratory tract. Apnea episodes with and without respiratory chest movement were classified as OSA and central sleep apnea, respectively. Mixed apnea was defined as episodes without respiratory effort and airflow in the first part of the event and with respiratory effort but without airflow in the last part of the event. Reduction in airflow by ≥30% for ≥10 s and leading to a ≥4% decrease in hemoglobin oxygen saturation (SpO_2_) was classified as hypopnea. Sleep-disordered breathing (SDB) was diagnosed if the rate of apnea or hypopnea per hour of recording (respiratory event index (REI)) exceeded four. SDB with REI of 5–14 events per hour (events/h) were classified as mild, 15–30 events/h as moderate, and >30 events/h as severe.

### 2.4. Statistical Analysis

The results were analyzed using MedCalc Version 20.106 (MedCalc Software Ltd., Ostend, Belgium). Depending on the normality of the distribution assessed by the Kolmogorov–Smirnov test, the quantitative parameters were presented as the arithmetic mean and standard deviation (SD) or the median and interquartile range (IQR). Qualitative data were expressed as numbers and percentages. Differences in the frequency and size of anthropometric and HSAT parameters by OSA severity and sex were calculated using the chi-square test, the student’s *t*-test for independent variables, or the Mann–Whitney U test, as required. Differences in the frequency and size of risk factors by OSA severity and sex were calculated using the chi-square test with the Cochran–Armitage test for trend or the Kruskal–Wallis test with the Jonckheere–Terpstra test for trend. The sensitivity, specificity, and positive and negative predictive values of the OSA questionnaires in the study group were calculated. The accuracy was calculated from the formula: the number of true positive cases (correctly identified as patient) plus the number of true negative cases (correctly identified as healthy) and divided by the number of all participants, expressed in percentages. A *p*-value less than 0.05 was considered statistically significant.

### 2.5. Ethical Considerations

The study was approved by the Bioethical Committee of the Medical University of Silesia in Katowice, Poland. Informed consent was obtained from all subjects involved in the study.

## 3. Results

During the study, 472 patients (Caucasians aged > 18 years) participated in CR. We collected complete medical documentation from 438 patients, including 349 (79.7%) men who ultimately constituted the study group. The mean age of participants was 59.92 ± 10.92 years. The anthropometric parameters and comorbidities of the study group are presented in Table 2.

In the study group of 438 patients, we assessed the OSA risk based on questionnaires. The high risk of OSA was calculated: in 248 (56.6%) participants (211 (48.2%) men and 37 (8.4%) women, *p* < 0.001) based on the STOP-BANG; in 163 (37.5%) participants (149 (34.3%) men and 14 (3.2%) women, *p* < 0.001) based on the ANC; in 115 (26.3%) participants (113 (25.8%) men and 2 (0.5%) women, *p* < 0.001) based on the 4-V; and in 45 (10.3%) participants (36 (8.2%) men and 9 (2.1%) women, *p* = 0.955) based on the ESS. Considering the results of all four questionnaires, a high risk of OSA was in 283 (64.6%) surveyed patients—240 (68.8%) men and 43 (48.3%) women, *p* < 0.001.

The technically adequate HSAT was performed on 275 patients—238 men and 37 women—who provided informed consent for further evaluation, irrespective of the estimated OSA risk. A total of 72 (26.2%) patients were not diagnosed with SDB. OSA was confirmed in 186 (68.0%) respondents; it was mild in 74 (31.1%) men and 11 (29.7%) women, moderate in 50 (21.0%) men and 3 (8.1%) women, and severe in 45 (18.9%) men and 3 (8.1%) women. In the study group, there were also 14 (5.1%) patients with central sleep apnea and 3 (1.1%) with mixed sleep apnea. We excluded these 17 patients from further analysis. Table 3 presents the characteristics of the HSAT parameters of patients with OSA according to sex.

In the group with moderate OSA, the median REI was 19.90 (IQR 17.25–24.08) events/h, while the percentage of total sleep time with oxyhemoglobin saturation below 90%, mean, and minimum SpO_2_ during sleep was 2.67% (0.71–8.59), 92.95 ± 1.73%, and 83.36 ± 5.54%, respectively. On the other hand, in the group with severe OSA, the median REI was 41.40 (33.90–57.10) events/h, the percentage of total sleep time with oxyhemoglobin saturation below 90% was 7.16% (1.80–23.39), the mean SpO_2_ was 92.11 ± 2.05%, and the minimum SpO_2_ was 80.87 ± 6.16%.

In turn, Table 4 shows the prevalence of risk factors and the score on the risk scales in the entire study group and in the subgroups, depending on the severity of OSA determined based on HSAT and sex.

The male-to-female prevalence ratio for OSA was 1.5:1. In men, the prevalence of risk factors such as age > 50 years, BMI > 35 kg/m^2^, high neck circumference, high blood pressure, snoring, and observed sleep apnea episodes increased proportionally to the severity of OSA. In women, this only concerned the prevalence of large neck circumference, BMI > 35 kg/m^2^, and awakenings from sleep. On the other hand, the value of the STOP-BANG and the ANC scores increased proportionally to the severity of OSA in both sexes, while the value of 4-V increased in this manner only in men. Tiredness/sleepiness, both analyzed as risk factors or counted as a score on the ESS, did not correlate with the severity of OSA. Tiredness/sleepiness were significantly more often reported by women than by men, *p* < 0.001.

The sensitivity, specificity, and positive and negative predictive values of the questionnaires were assessed correspondingly for both sexes due to the small size of the female subgroup. The results are presented in Table 5.

## 4. Discussion

In the presented analysis, the authors confirmed the high prevalence of OSA risk factors in middle-aged and older patients after MI participated in ambulatory CR. That prevalence is reflected in the risk questionnaire results (high risk of OSA in 64.6% of surveyed) and confirmed in the HSAT (36.7% of patients with moderate-to-severe OSA). The questionnaires based on a combination of symptoms and risk factors, in particular the STOP-BANG and the ANC, were more sensitive in predicting moderate-to-severe OSA. The ESS, based solely on assessing pathological daytime sleepiness, was the least sensitive. The specificity of the ANC and the 4-V exceeded 60% and were much higher than that of the STOP-BANG. The ANC had both sensitivity and specificity exceeding 60% in predicting moderate-to-severe OSA and was the only scale with a positive predictive value exceeding 50%. Thus, the ANC seems to be the most useful in post-MI patients in clinical practice.

In Poland, the prevalence of CAD is about 4.0% higher than the European Union average [40,41]. The mortality rate of patients with MI is 8.4% during hospitalization and another 9.8% in the one-year follow-up after discharge from the hospital [42]. Intensive secondary prevention focused on lifestyle changes, CR, and the reduction in all known CV risk factors reduces the risk of subsequent CV events [43,44,45]. On the other hand, the prevalence of moderate-to-severe OSA exceeds 15 to 35% among middle-aged and older adults in the general population (17.8% in Poland) [1,46] and 35% to 65% in post-MI patients [1,47,48,49]. In our group of patients from the MC-AMI program, moderate-to-severe OSA was found in 36.7% of respondents. As recommended by scientific societies [1,50], treatment of all patients with OSA and CAD should be considered. Treatment includes lifestyle changes, weight loss, and positive airway pressure therapy. Therefore, performing screening tests such as a targeted medical history, risk questionnaires, or using sleep apnea screening devices in the post-MI patient referred for CR seems particularly justified [1,50].

The OSA risk questionnaires have been validated in the general population for men and women. These scales’ sensitivity and specificity (REI ≥ 15 events/h) have been assessed as 87% and 43% for the STOP-BANG; 63% and 72% for the ANC; 24% and 93% for the 4-V; and 39% and 71% for the ESS [29,34,35,36,37,38]. These questionnaires are characterized by high accuracy in the general population, while in the cardiac population, the sensitivity and specificity of some of them are much lower. Hwang et al. [51], in a literature search of 1894 patients with CV risk factors, revealed the STOP-BANG specificity for moderate-to-severe OSA to be 22.5%, resulting in a high rate of false positives. An even lower STOP-BANG specificity of 13.0% was reported by Reuter et al. [52] and Nunes et al. [53]. On the other hand, the sensitivity of the ESS (probably the most widely known SDB risk scale in Poland) seems insufficient, which limits this scale’s usefulness in the risk assessment and qualification for OSA treatment in post-MI population. Hupin et al. [28] and Glantz et al. [49], in a revascularized CAD cohort, found the absence of typical daily OSA symptoms in 61.8–65.0% of patients. The low level of daytime sleepiness in the population with CV diseases is also reported by other authors [52,53]. In our sample, 36.7% of patients, mainly men, had moderate-to-severe OSA, and only 16.8% with such a risk scored more than ten points on the ESS. Conversely, excessive sleepiness in patients already diagnosed with OSA is considered to be a factor in increasing the risk of CV events [54]. The 4-V scale’s sensitivity, primarily validated for Japan’s population by Takegami et al. [38], is lower for Europe [35] and the United States [34], with high specificity maintained. In our sample from the Polish post-MI group, its sensitivity reached only 45.54%. There is a lack of literature on the accuracy of ANC in the post-MI patient. However, based on our study, we believe that this scale, with a sensitivity and specificity of 61%, may help identify moderate-to-severe OSA in such patients.

Another problem may be in underestimating the risk of OSA in women [55,56]. In most epidemiological reports, the prevalence of OSA is higher in men than in women, with a male-to-female ratio of approximately 1.5:1 to 3:1 [55,56]. These statistics may be an underestimate due to less specific OSA symptoms reported by women (fatigue, morning headache, and difficulty initiating sleep rather than snoring and apnea) [21,56,57,58]. Furthermore, the OSA risk calculated based on risk scales is usually higher for men, as it assumes additional points for the male sex (STOP-BANG, 4-V) [36,38] or determines the same threshold of pathological neck circumference for both sexes (STOP-BANG, ANC) [36,37], which may translate into a lower sensitivity but higher specificity of some risk scales, e.g., STOP-BANG in women [59]. Some authors report [60] that anthropometric measurements, such as the neck, waist, and hip circumferences, correlate less well with the severity of OSA in male patients than in females. In our group, OSA’s prevalence was only 1.5 times higher in men than women. The small difference between men and women may be due to the higher incidence of OSA in women post-MI than in the general population and a higher percentage of diagnoses in women without typical symptoms, the absence of which did not limit HSAT’s performance in our study. Unfortunately, the female group in the studied population was too small to assess the sensitivity and specificity of risk questionnaires separately for both sexes.

Last but not least, it is worth remembering that post-MI patients are cared for by cardiologists who rarely deal with SDB in their daily practice. The user-friendly and not time-consuming risk scales can encourage OSA diagnostics in cardiological outpatient and inpatient clinics or during early CR after MI. Therefore, it is worth considering ANC, which can be performed at the patient’s bedside by counting four simple parameters. Both the sensitivity and specificity of this scale for the studied population were high enough and totaled to 61%. Patients with CV diseases and a high risk of OSA should be referred for further diagnostics [1] at the sleep laboratory (reimbursed service in Poland) or for HSAT, provided that its availability improves (the outpatient service is not currently reimbursed by the National Health Fund, the obligatory health insurer in Poland).

### Study Limitations

Risk scales and questionnaires analyzed in our study are the only auxiliary tool in SDB diagnosis [1,29]. HSAT was used to diagnose OSA instead of the gold standard, the level 1 polysomnography performed in the sleep laboratory [29]. However, HSAT is often employed in research for examining the relationship between OSA and differential CV diseases [61,62,63,64] and is recommended for screening in CR [50]. We repeated or rejected all inconclusive or technically inadequate results to ensure the credibility of the research.

Due to the lower prevalence of CAD and OSA in women, the female group in the studied population was relatively small, which could affect the statistical significance of the analysis. Similarly, this group was too small to assess the sensitivity and specificity of risk questionnaires separately for both sexes.

Patients with symptomatic heart failure and significantly reduced left ventricular ejection fraction are not eligible for outpatient CR, hence the low percentage of patients with central sleep apnea in the study sample. However, such pre-selection in the case of OSA risk assessment by HSAT is an asset rather than a limitation of the study. Likewise, patients with severe or multiple comorbidities were not referred to the day treatment rehabilitation ward but the stationary CR ward. Thus, the patients with severe chronic obstructive pulmonary disease, heart failure in NYHA III/IV class, tachyarrhythmias, etc., may have been underrepresented in the study group. According to data from the SILesian CARDiovascular (SILCARD) registry [32], in 2017–2018 in the Silesian Voivodeship, Poland, 36% of MC-AMI patients participated in ambulatory CR.

The presented results are not derived from a clinical trial, and no clinical outcomes related to the frequency of OSA diagnoses have been studied.

## 5. Conclusions

OSA is a common CV risk factor among post-MI patients. The ANC, with a sensitivity and specificity of 61% in predicting moderate-to-severe OSA, appears to be the most helpful risk questionnaire in clinical practice. The ESS sensitivity of only 16.83% is insufficient to assess the risk of OSA in the post-MI patient population.

## Figures and Tables

**Table 1 jpm-13-00642-t001:** Parameters taken into account in the risk assessment of obstructive sleep apnea on individual scales. ANC: the adjusted neck circumference; ESS: the Epworth sleepiness scale; STOP-BANG: the STOP-BANG questionnaire; 4-V: the 4-variable screening tool.

Symptoms/Risk Factors	Risk Questionnaires			
	STOP-BANG	4-V	ESS	ANC
Age older than 50 years	+			
Male sex	+	+		
Increased body mass index	+	+		
Neck circumference	+			+
Lond and frequent snoring	+	+		+
Observed daytime sleepiness	+			+
Excessive daytime sleepiness	+		+	
High blood pressure/hypertension	+	+		+

**Table 2 jpm-13-00642-t002:** Characteristics of anthropometric parameters and comorbidities in the surveyed men and women.

All Surveyed	Male	Female	*p* Value
	*n* = 349 (79.7%)	*n* = 89 (20.3%)	
	Anthropometric parameters		
Age (years), mean ± SD	59.76 ± 11.08	60.55 ± 10.31	0.545
Neck circumference (cm), median (IQR)	43.0 (41.0–44.5)	38.0 (36.0–39.0)	<0.001
Body weight (kg), mean ± SD	88.85 ± 14.77	74.84 ± 14.48	<0.001
BMI (kg/m^2^), mean ± SD	29.05 ± 4.42	28.55 ± 5.03	0.354
	Comorbidities		
CAD with a history of MI, *n* (%)	349 (100.0%)	89 (100.0%)	1.000
Hypertension, *n* (%)	282 (80.8%)	77 (86.5%)	0.211
Atrial fibrillation, *n* (%)	26 (7.4%)	1 (1.1%)	0.027
History of ischemic stroke, *n* (%)	7 (2.0%)	1 (1.1%)	0.580
HFmrEF, *n* (%)	73 (21.3%)	18 (20.7%)	0.894
HFrEF, *n* (%)	7 (2.0%)	2 (2.2%)	0.886
LVEF (%), mean ± SD	52.98 ± 5.76	53.64 ± 6.10	0.341
Diabetes, *n* (%)	78 (22.3%)	22 (24.7%)	0.635
COPD, *n* (%)	16 (4.6%)	2 (2.2%)	0.322
Smoking status (within the last 5 years), *n* (%)	138 (39.5%)	45 (50.6%)	0.060

BMI: body mass index; CAD: coronary artery disease, cm: centimeter, COPD: chronic obstructive pulmonary disease, HFmrEF: heart failure with mildly reduced ejection fraction, HFrEF: heart failure with reduced ejection fraction, IQR: interquartile range, LVEF: left ventricular ejection fraction, MI: myocardial infarction, SD: standard deviation.

**Table 3 jpm-13-00642-t003:** Characteristics of polysomnographic parameters in men and women with obstructive sleep apnea.

Polysomnographic Parameters	Male	Female	*p* Value
	*n* = 238	*n* = 37	
Total sleep time (min), mean ± SD	463.64 ± 112.85	481.89 ± 136.08	0.546
REI (events/h), median (IQR)	12.75 (5.80–25.30)	6.50 (3.43–13.23)	0.007
Average episode duration (s), mean ± SD	22.93 ± 6.37	20.47± 5.96	0.029
Max episode duration (s), mean ± SD	63.14 ± 28.73	47.89 ± 27.82	0.003
Average SpO_2_ (%), mean ± SD	92.79 ± 1.85	92.92 ± 2.60	0.771
Minimal SpO_2_ (%), mean ± SD	83.46 ± 6.32	85.95 ± 4.27	0.003
TST 90 (%), median (IQR)	1.80 (0.27–7.58)	1.40 (0.00–10.30)	0.550
Mean HR at night, mean ± SD	59.35 ± 7.79	59.79 ± 7.44	0.754
Minimal HR at night, mean ± SD	46.78 ± 7.27	49.47 ± 7.03	0.044

HR: heart rate measured by pulse oximetry; IQR: interquartile range; OSA: obstructive sleep apnea; REI: respiratory event index; SpO_2_: arterial oxygen saturation estimated by pulse oximetry; SD: standard deviation; TST 90: the percentage of total sleep time with oxyhemoglobin saturation below 90%.

**Table 4 jpm-13-00642-t004:** The prevalence of risk factors and score on the risk scales in the entire study group and subgroups, depending on the severity of obstructive sleep apnea and sex.

	Sex	All Surveyed *n* = 438	None SDB *n* = 72	Mild OSA *n* = 85	Moderate OSA *n* = 53	Severe OSA *n* = 48	*p* Value for Trend
Age >50 years, *n* (%)	M	276 (79.1%)	37 (66.1%)	57 (77.0%)	39 (78.0%)	40 (88.9%)	0.010
	F	73 (82.0%)	13 (81.2%)	10 (90.9%)	3 (100.0%)	3 (100.0%)	0.225
BMI > 35 kg/m^2^, *n* (%)	M	34 (9.7%)	5 (8.9%)	5 (6.8%)	9 (18.0%)	9 (20.0%)	0.028
	F	8 (9.0%)	0 (0.0%)	5 (45.5%)	0 (0.0%)	2 (66.7%)	0.014
Neck circumference > 40 cm, *n* (%)	M	272 (77.9%)	38 (67.9%)	62 (83.8%)	42 (84.0%)	42 (93.3%)	0.002
	F	14 (15.7%)	0 (0.0%)	5 (45.5%)	2 (66.7%)	1 (33.3%)	0.014
BP ≥ 140/90 mmHg/hypertension, *n* (%)	M	282 (80.8%)	42 (75.0%)	60 (81.1%)	43 (86.0%)	41 (91.1%)	0.025
	F	77 (86.5%)	14 (87.5%)	11 (100.0%)	2 (66.7%)	3 (100.0%)	0.816
Loud and frequent snoring, *n* (%)	M	160 (45.8%)	18 (32.1%)	44 (59.5%)	30 (60.0%)	34 (75.6%)	<0.001
	F	42 (47.2%)	8 (50.0%)	5 (45.5%)	1 (33.3%)	3 (100.0%)	0.337
Observed stop breathing/choking/gasping, *n* (%)	M	74 (21.3%)	10 (17.9%)	21 (28.4%)	19 (38.0%)	15 (33.3%)	0.041
	F	14 (15.7%)	2 (12.5%)	3 (27.3%)	0 (0.0%)	3 (100.0%)	0.014
Daytime sleepiness/tiredness, *n* (%)	M	133 (38.1%)	27 (48.2%)	28 (37.8%)	19 (38.0%)	26 (57.8%)	0.401
	F	60 (67.4%)	13 (81.2%)	9 (81.8%)	1 (33.3%)	3 (100.0%)	0.827
Sleepiness on ESS (pts), median (IQR)	M	4.0	4.0	5.0	5.0	6.0	0.111
		(2.0–8.0)	(3.0–7.5)	(2.0–8.0)	(4.0–9.0)	(3.0–8.5)	
	F	4.0	5.0	7.0	4.0	3.0	0.467
		(2.0–8.0)	(2.0–7.5)	(3.8–10.0)	(3.3–7.8)	(1.5–14.3)	
STOP-BANG (pts), median (IQR)	M	5.0	4.0	5.0	5.0	6.0	<0.001
		(4.0–5.0)	(3.0–5.0)	(4.0–6.0)	(4.0–6.0)	(5.0–6.0)	
	F	3.0	3.0	4.0	3.0	6.0	0.003
		(2.0–4.0)	(3.0–3.5)	(3.3–5.0)	(1.5–4.5)	(6.0–6.0)	
4-V (pts), median (IQR)	M	11.0	10.0	13.0	13.0	14.0	<0.001
		(9.0–14.0)	(9.5–13.5)	(10.0–14.0)	(10.0–15.0)	(11.0–15.0)	
	F	7.0	7.0	9.0	7.0	11.0	0.050
		(6.0–10.0)	(6.0–9.5)	(7.0–11.0)	(6.3–10.8)	(11.0–11.0)	
ANC (cm), median (IQR)	M	47.5	46.0	48.0	49.0	50.0	<0.001
		(44.5–50.5)	(43.3–50.0)	(45.0–50.5)	(47.0–53.0)	(47.0–54.0)	
	F	43.0	41.0	46.0	45.0	50.0	<0.001
		(40.0–46.0)	(37.8–42.5)	(44.3–49.8)	(42.8–48.0)	(49.3–50.8)	

ANC: the adjusted neck circumference, BMI: body mass index, BP: blood pressure, cm: centimeter, ESS: the Epworth sleepiness scale, F: female, IQR: interquartile range, M: male; No: number, SDB: sleep-disordered breathing, STOP-BANG: the STOP-BANG questionnaire, 4-V: the 4-variable screening tool.

**Table 5 jpm-13-00642-t005:** The sensitivity, specificity, and positive and negative predictive values of the questionnaires in the prediction of moderate-to-severe obstructive sleep apnea in the study population.

Questionnaire	Sensitivity (%) *	Specificity (%) *	PPV (%) *	NPV (%) *	Accuracy (%)
STOP-BANG	79.21 (70.0–86.6)	35.67 (28.2–43.7)	44.2 (40.5–48.0)	72.7 (63.3–80.5)	52.71
ANC	61.39 (51.2–70.9)	61.15 (53.1–68.8)	50.4 (44.2–56.6)	71.1 (65.1–76.4)	61.24
4-V	45.54 (35.6–55.8)	68.79 (60.9–75.9)	48.4 (40.6–56.3)	66.3 (61.5–70.7)	60.01
ESS	16.83 (10.1–25.6)	87.90 (81.7–92.6)	47.2 (32.8–62.1)	62.2 (59.7–64.6)	59.30

* Data presented as average (95% confidence interval); ANC: the adjusted neck circumference; ESS: the Epworth sleepiness scale; NPV: negative predictive value; PPV: positive predictive value; STOP-BANG: the STOP-BANG questionnaire; 4-V: the 4-variable screening tool.

## Data Availability

The data presented in this study are available on request from the Department of Electrocardiology and Heart Failure, Medical University of Silesia in Katowice (Poland). The data are not publicly available due to privacy restrictions.

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
