# Peer review of "The Usefulness of Questionnaires in Assessing the Risk of Obstructive Sleep Apnea in Patients in the Managed Care after Acute Myocardial Infarction Program—The Results of a Cross-Sectional Study"

_jpm, 2023, doi:10.3390/jpm13040642_

Round 1
Reviewer 1 Report
This is a simple but well-conducted and interesting study testing the usefulness of various questionnaires in assessing and screening obstructive sleep apnea in patients with recent MI. During post-MI rehabilitation, diagnosing and treating COPD and sleep apnea may bring better long-term outcomes. Oftentimes, this remains underdiagnosed and the authors aimed to address this gap. Their rate of diagnosis OSA in their cohort was high, over 60%.
Specific comments:
- Abstract: clear and well written.
- Introduction: well-written. Maybe put a phrase on how OSA is related to obesity and COPD and these entities oftentimes cause HFpEF, which worsens the prognosis of these patients by additionally aggravating the stiffness and the filling pressures of the left ventricle.
- Methods: "the inclusion criteria for the MC-AMI program, described in detail elsewhere" - maybe write just 1 phrase what was the main principle behind this program, also specify what MI was included (NSTEMI, STEMI, both, late-ACS with reduced EF, etc).
- Methods: were there any exclusion criteria? Like other associated lung pathologies?
- Results: very clear.
- Discussion: the mean BMI and weight surprisingly was not that high. Also, the rate of COPD and AF was low, only HTN was prevalent. Maybe comment on that. Could a hidden/underdiagnosed HFpEF aggravate OSA?
- Discussion: comment that untreated moderate-severe OSA is independently associated with a significantly increased risk of repeat revascularization after PCI and PTCA in lower limbs and atherosclerosis may be aggravated by an unhealthy regime and systemic metabolic diseases with OSA being the "peak of the iceberg". Please cite:
Bhama JK, Spagnolo S, Alexander EP, Greenberg M, Trachiotis GD. Coronary revascularization in patients with obstructive sleep apnea syndrome. Heart Surg Forum. 2006;9(6):E813-7. doi: 10.1532/HSF98.20061072.
Achim A, Stanek A, Homorodean C, Spinu M, Onea HL, Lazăr L, Marc M, Ruzsa Z, Olinic DM. Approaches to Peripheral Artery Disease in Diabetes: Are There Any Differences? Int J Environ Res Public Health. 2022 Aug 9;19(16):9801. doi: 10.3390/ijerph19169801.
Limitations: state that this was not a clinical study and no clinical outcomes following the rate of OSA diagnosis were studied.
Conclusion: clear.
Author Response
Dear Reviewer,
Thank you for your kind evaluation of our work. We hope that we have been able to answer your questions exhaustively.
Comments and Suggestions for Authors
- Abstract: clear and well written.
Reply: Thank you for accepting the section.
- Introduction: well-written. Maybe put a phrase on how OSA is related to obesity and COPD and these entities oftentimes cause HFpEF, which worsens the prognosis of these patients by additionally aggravating the stiffness and the filling pressures of the left ventricle.
Reply: Thank you for this remark.
OSA-induced periodic hypoxia through activation of the sympathetic nervous system and the renin-angiotensin-aldosterone system leads to peripheral vasoconstriction, sodium and water retention, resistant hypertension, LV hypertrophy, and progression of LV diastolic dysfunction.* In addition, oxidative stress, endothelial dysfunction, systemic inflammation, and OSA-related metabolic disorders promote atherogenesis, and arterial stiffness.** Thus OSA can be considered a risk factor for heart failure with preserved ejection fraction (HFpEF). Some other comorbidities, such as arterial hypertension, chronic kidney disease, chronic obstructive pulmonary disease, and primarily obesity with concomitant metabolic disorders or diabetes, contribute to the development of HFpEF with coronary microvascular dysfunction, myopathy, and atrial and ventricular fibrosis.*** We added some information about HFpEF and its risk factors to the Introduction section.
*Sanderson JE, Fang F, Lu M, Ma CY, Wei YX. Obstructive sleep apnoea, intermittent hypoxia and heart failure with a preserved ejection fraction. Heart 2021;107:190–194.
** Drager LF, Polotsky VY, Lorenzi-Filho G. Obstructive sleep apnea: an emerging risk factor for atherosclerosis. Chest 2011;140:534–542.
Achim A, Stanek A, Homorodean C, Spinu M, Onea HL, Lazăr L, Marc M, Ruzsa Z, Olinic DM. Approaches to Peripheral Artery Disease in Diabetes: Are There Any Differences? Int J Environ Res Public Health. 2022;19(16):9801.
*** Lin Y, Fu S, Yao Y, Li Y, Zhao Y, Luo L. Heart failure with preserved ejection fraction based on aging and comorbidities. J Transl Med. 2021;19(1):291.
- Methods: "the inclusion criteria for the MC-AMI program, described in detail elsewhere" - maybe write just 1 phrase what was the main principle behind this program, also specify what MI was included (NSTEMI, STEMI, both, late-ACS with reduced EF, etc).
Reply: Thank you for this remark. Following the reviewer's suggestion, we have extended the description of the inclusion criteria for the MC-AMI program.
The MC-AMI program covers all patients hospitalized for acute myocardial infarction (both ST-elevation MI and non-ST-elevation MI) diagnosed in line with the Fourth Universal Definition of Myocardial Infarction.* The MC-AMI program provides treatment in four treatment modules: Module I - the treatment of the acute phase of MI (i.e., angioplasty, arterial bypass grafting, follow-up visit within 14 days after discharge); Module II - cardiac rehabilitation; Module III - electrotherapy (i.e., implantation of an implantable cardiac defibrillator in primary prevention of sudden cardiac death, if necessary); and Module IV - specialized cardiac care during the 12 months following MI.
*Thygesen K, Alpert JS, Jaffe AS, Chaitman BR, Bax JJ, Morrow DA, White HD; Executive Group on behalf of the Joint European Society of Cardiology (ESC)/American College of Cardiology (ACC)/American Heart Association (AHA)/World Heart Federation (WHF) Task Force for the Universal Definition of Myocardial Infarction. Fourth Universal Definition of Myocardial Infarction (2018). J Am Coll Cardiol. 2018;72(18):2231-2264.
- Methods: were there any exclusion criteria? Like other associated lung pathologies?
Reply: Thank you for this question.
The only criterion for exclusion from the study was the current treatment of OSA with positive airway pressure or intraoral devices. None of the comorbidities was a criterion for exclusion from the study. However, patients with severe or multiple comorbidities were not referred to the Day Rehabilitation Department but the stationary ward and therefore are underrepresented in the study group.
We added this information to the limitations of the study.
- Results: very clear.
Reply: Thank you for accepting the section.
- Discussion: the mean BMI and weight surprisingly was not that high. Also, the rate of COPD and AF was low, only HTN was prevalent. Maybe comment on that. Could a hidden/underdiagnosed HFpEF aggravate OSA?
Reply: Thank you for this remark.
The presented data come from anthropometric measurements taken on admission to the rehabilitation department, data from the patients' interview, and diagnoses from medical records. Unfortunately, we do not have data on the prevalence of HFpEF in the study sample.
The relatively low percentage of obese people with numerous comorbidities can be explained as follows: 1) The study sample consists of middle-aged and older patients, including 67.0% of men and 40.9% of women of working age (pre-retirement), ranging from 27 years, 2) Patients had to be suitable for rehabilitation in outpatients settings according to the A-C model, which excluded patients with severe or multiple comorbidities - such patients were not referred to the Day Treatment Rehabilitation Word but rehabilitated in the inpatient setting. According to data from the SILesian CARDiovascular (SILCARD) registry,* in 2017-2018 in the Silesian Voivodeship, Poland, 36% of MC-AMI patients participated in ambulatory CR.
We added this information to the limitations of the study.
*Wita K, Kułach A, Sikora J, Fluder J, Nowalany-Kozielska E, Milewski K, Pączek P, Sobocik H, Olender J, Szela L, Kalarus Z, Buszman P, Jankowski P, Gąsior M. Managed Care after Acute Myocardial Infarction (MC-AMI) Reduces Total Mortality in 12-Month Follow-Up-Results from a Poland's National Health Fund Program of Comprehensive Post-MI Care-A Population-Wide Analysis. J Clin Med. 2020;9(10):3178.
- Discussion: comment that untreated moderate-severe OSA is independently associated with a significantly increased risk of repeat revascularization after PCI and PTCA in lower limbs and atherosclerosis may be aggravated by an unhealthy regime and systemic metabolic diseases with OSA being the "peak of the iceberg". Please cite:
Bhama JK, Spagnolo S, Alexander EP, Greenberg M, Trachiotis GD. Coronary revascularization in patients with obstructive sleep apnea syndrome. Heart Surg Forum. 2006;9(6):E813-7. doi: 10.1532/HSF98.20061072.
Achim A, Stanek A, Homorodean C, Spinu M, Onea HL, Lazăr L, Marc M, Ruzsa Z, Olinic DM. Approaches to Peripheral Artery Disease in Diabetes: Are There Any Differences? Int J Environ Res Public Health. 2022 Aug 9;19(16):9801. doi: 10.3390/ijerph19169801.
Reply: Thank you for your this comment and for pointing out a noteworthy reference.
We have added information on OSA as a prognostic factor in surgically revascularized patients to the Introduction section (Bhama et al.). We also extended the description of pathogenetic factors in patients with OSA affecting the development of cardiovascular diseases, taking into account the progression of peripheral artery atherosclerosis and arterial stiffness as a consequence of OSA (Achim et al.).
Limitations: state that this was not a clinical study and no clinical outcomes following the rate of OSA diagnosis were studied.
Reply: Thank you for your this comment. This information has been added to the limitations of the study.
Conclusion: clear.
Reply: Thank you for accepting the section.
Corresponding author,
Danuta Loboda, MD, PhD
Reviewer 2 Report
Dear authors,
I have studied with great interest the manuscript “The Usefulness of Questionnaires in Assessing the Risk of Obstructive Sleep Apnea in Patients in the Managed Care after Acute Myocardial Infarction Program – the Results of a Cross-Sectional Study”.
The main question is addressed by the research was to evaluate the prevalence of OSA in patients after MI using various questionnaires and to estimate their usefulness. The authors showed that OSA is common in post-MI patients. The ANC most accurately estimates the risk of OSA eligible for positive airway pressure therapy. The sensitivity of the ESS in the post-MI population is insufficient and limits this scale's usefulness in risk assessment and qualification for treatment.
The manuscript is clearly exposed and well written, the text clear and easy to read. The topic is original. The figures and tables correspond to the description in the text, are well designed and reflect important information. The introduction, methods, result, discussion sections are well-structured and includes appropriate information.
But I have some comments to improve the quality of the presentation.
1. The authors said that the technically adequate HSAT was performed on 275 patients who gave who gave informed consent for further evaluation, irrespective of the estimated OSA risk. But, if all those patients were high-risk, perhaps this explains such a high prevalence of OSA in the investigated population?
2. The authors provided the sensitivity, specificity, and positive and negative predictive value of the questionnaires. Please, add accuracy in the table 4.
But generally, I express my gratitude to the authors for their great work done. These results could open the door to the diagnostic strategies that could improve the quality of life of affected patients.
Author Response
Dear Reviewer,
Thank you for your kind evaluation of our work. We hope that we have been able to answer your questions exhaustively.
Comments and Suggestions for Authors
- The authors said that the technically adequate HSAT was performed on 275 patients who gave who gave informed consent for further evaluation, irrespective of the estimated OSA risk. But, if all those patients were high-risk, perhaps this explains such a high prevalence of OSA in the investigated population?
Reply: Thank you for your this comment.
Patients with coronary artery disease (CAD) are at high risk for OSA complications [1-9]. However, not everyone is at high risk of OSA as assessed by the risk scales. Due to the planned assessment of the sensitivity and specificity of the risk scales, we performed the HSAT on all patients who consented to such a test, regardless of the score obtained in the risk scales - from low to high.
The prevalence of moderate-to-severe OSA (AHI >15 events/hour) exceeds 15 to 35% among middle-aged and older adults in the general population (17.8% in Poland) [1,36] and 35% to 65% in post-MI patients [1,37-39]. This results, among others, from similar risk factors for OSA and CAD, such as older age, male sex, obesity, and metabolic syndrome [19].* The prevalence of OSA with REI >15 events/hour in our group is 36.7% - it is twice as high as the average rate of moderate-to-severe OSA in the general population in Poland but similar to the average for patients with MI.
*Peppard PE, Young T, Barnet JH, et al. Increased prevalence of sleep-disordered breathing in adults. Am J Epidemiol. 2013; 177(9): 1006–1014.
Abe H, Takahashi M, Yaegashi H, et al. Efficacy of continuous positive airway pressure on arrhythmias in obstructive sleep apnea patients. Heart Vessels. 2010; 25(1): 63–69.
Szymański FM, Płatek AE, Karpiński G, et al. Obstructive sleep apnoea in patients with atrial fibrillation: prevalence, determinants and clinical characteristics of patients in Polish population. Kardiol Pol. 2014; 72(8): 716–724.
Pływaczewski R, Bieleń P, Bednarek M, et al. Influence of neck circumference and body mass index on obstructive sleep apnoea severity in males [article in Polish]. Pneumonol Alergol Pol. 2008; 76(5): 313–320.
- The authors provided the sensitivity, specificity, and positive and negative predictive value of the questionnaires. Please, add accuracy in the table 4.
Reply: Thank you for this remark.
The accuracy of the STOP-BANG questionnaire was 52.71%, the Adjusted Neck Circumference 61.24%, the Epworth Sleepiness Scale 59.30%, and the 4-Variable Screening Tool 60.01%.
This information has been added to Table 4.
Corresponding author,
Danuta Loboda, MD, PhD
Round 2
Reviewer 2 Report
The authors have adressed all my comments